PREPARED FOR SUBMISSION TO SCIPOST

# Large $N$ instantons from topological strings

**Marcos Mariño and Ramon Miravitllas**

*Département de Physique Théorique et Section de Mathématiques*
*Université de Genève, Genève, CH-1211 Switzerland*

*E-mail:* Marcos.Marino@unige.ch, Ramon.MiravitllasMas@unige.ch

ABSTRACT: The $1/N$ expansion of matrix models is asymptotic, and it requires non-perturbative corrections due to large $N$ instantons. Explicit expressions for large $N$ instanton amplitudes are known in the case of Hermitian matrix models with one cut, but not in the multi-cut case. We show that the recent exact results on topological string instanton amplitudes provide the non-perturbative contributions of large $N$ instantons in generic multi-cut, Hermitian matrix models. We present a detailed test in the case of the cubic matrix model by considering the asymptotics of its $1/N$ expansion, which we obtain at relatively high genus for a generic two-cut background. These results can be extended to certain non-conventional matrix models which admit a topological string theory description. As an application, we determine the large $N$ instanton corrections for the free energy of ABJM theory on the three-sphere, which correspond to D-brane instanton corrections in superstring theory. We also illustrate the applications of topological string instantons in a more mathematical setting by considering orbifold Gromov–Witten invariants. By focusing on the example of $\mathbb{C}^3/\mathbb{Z}_3$, we show that they grow doubly-factorially with the genus and we obtain and test explicit asymptotic formulae for them.

## 1   Introduction

In spite of its arcane nature, topological string theory on Calabi–Yau (CY) manifolds has been extremely useful in addressing more mundane problems. Originally [1], topological strings were constructed as physical counterparts of Gromov–Witten theory, and physics-inspired results in topological string theory have had an enormous impact on algebraic geometry. It was later understood in [2] that matrix models are in a sense a special case of topological string theory. This opened the way to solve some important but difficult matrix models by using topological string ideas. A remarkable example is the matrix model describing the free energy of ABJM theory [3] on the three-sphere [4], which was solved in the $1/N$ expansion in [5, 6] by using topological string theory on a non-compact CY manifold.

Perturbative topological string theory is relatively well understood, and it has provided most of the applications that we have just mentioned. One of the most important tools in formulating and calculating the perturbative expansion of topological string theory is the BCOV holomorphic anomaly equations (HAE) [7, 8], which have been applied very successfully to both toric [9] and compact [10] CY manifolds. When matrix models are realized as topological strings, the perturbative string expansion corresponds to the $1/N$ expansion, which is governed as well by the HAE. This was first pointed out in [11], and then proved in [12] as a consequence of the topological recursion of [13].

The non-perturbative aspects of topological strings are less understood, and there are different schools of thought on how to deal with them. In [14] it was suggested to address this problem in a conservative way, by exploiting the well-known connection between non-perturbative sectors and the large order behavior of perturbation theory. This connection is the basis of the theory of resurgence [15–19], and in recent years many interesting results on topological string theory have been obtained by applying the tools and ideas of resurgence. In the pioneering papers [20, 21] it was proposed to use trans-series solutions to the HAE in order to obtain non-perturbative effects in topological string theory. This idea has been further developed recently, and as consequence exact formulae for multi-instanton amplitudes have been obtained both for local [22] and compact [23] CY manifolds.

It is natural to ask what are the implications of these new non-perturbative results for the $1/N$ expansion of matrix models. This expansion is known to be asymptotic, and therefore it is expected to have exponentially small, non-perturbative corrections, due to so-called large $N$ instantons (see [18, 24] for a detailed introduction). In the case of one-cut Hermitian matrix models, large $N$ instantons take the form of eigenvalue tunneling [25, 26]. Although this mechanism has been known for a long time, the first detailed calculation of multi-instanton amplitudes in one-cut Hermitian matrix models with polynomial potentials was only done in [27, 28] (see also [29] for a generalization to the two-matrix model case). The results in [27, 28] were tested by verifying that that the resulting amplitudes control the asymptotics of the $1/N$ expansion. However, in the case of general multi-cut matrix models, large $N$ instanton corrections are not fully understood. Naif expectations based on generalizations of eigenvalue tunneling fail to capture the asymptotic behavior of the $1/N$ expansion, as shown in [30].

In this paper we argue that the topological string instanton amplitudes obtained in [22, 23] provide the sought-for non-perturbative corrections due to large $N$ instantons of Hermitian multi-cut matrix models, at generic points in moduli space. This follows from the fact that the $1/N$ expansion is governed by the HAE of [8], and the instanton amplitudes of [22, 23] are derived based only on these equations and on boundary conditions which are also satisfied by matrix models. We test our claim in detail by considering the simplest two-cut matrix model, based on a cubic potential, and we show that the asymptotics of the $1/N$ expansion around generic two-cut saddle-points is controlled by the instanton amplitudes of [22, 23].

There are matrix models which are not of the conventional form but are closely related to topological string theory and governed by the HAE equations. These include Chern–Simons type matrix models, like the ones considered in [31, 32]. An important related example, as we mentioned above, is the ABJM matrix model, which was extensively studied in the context of the $\text{AdS}_4/\text{CFT}_3$ correspondence. Non-perturbative aspects of this model were discussed in [33], but precise large $N$ instanton amplitudes were not known. This is a particularly interesting issue since, as proposed in [33], some of these large $N$ instantons correspond to D-brane instantons in superstring theory. It is clear from the above that the large $N$ instantons of the ABJM matrix model should be also given by the topological string instanton amplitudes of [22, 23], and in this paper we test this in detail, completing in this way the picture developed in [33].

This work is focused on the applications of topological string instantons to large $N$ instantons of matrix models, but there are more mathematical applications of the results in [22, 23]. As an example of this type of applications, we also consider in this paper orbifold Gromov–Witten invariants, which have been studied in both algebraic geometry and topological string theory. We focus on the orbifold Gromov–Witten theory of $\mathbb{C}^3/\mathbb{Z}_3$, which is one of the most famous examples, and we show that these invariants grow doubly factorially with the genus at fixed degree, in contrast to conventional Gromov–Witten invariants [34]. In addition, we obtain explicit

and detailed formulae for their large genus asymptotics from the topological string instanton amplitudes of [22, 23].

This paper is organized as follows. In section 2 we briefly review the results on topological string instantons obtained in [22, 23], building on [20, 21]. In section 3 we consider the application to large $N$ instantons in multi-cut, Hermitian matrix models, and we present detailed tests in the two-cut, cubic matrix model. In section 4 we study large $N$ instantons in the ABJM matrix model. In section 5 we apply the results reviewed in section 2 to obtain the asymptotic behavior of orbifold Gromov–Witten invariants, in the case of $\mathbb{C}^3/\mathbb{Z}_3$. Finally, in section 6 we present our conclusions and some prospects for future developments. An Appendix includes some details on the parametrization of the moduli space of the cubic matrix model, used in section 3.

## 2 Instantons in topological string theory

In this section we briefly review the results on topological string instantons obtained in [22, 23], building on previous work in [20, 21, 35].

The basic quantities in topological string theory are the genus $g$ free energies $\mathcal{F}_g(t_a)$, where $t_a$, $a = 1, \cdots, n$, are flat coordinates which parametrize the moduli space of a CY threefold. In this paper we will restrict ourselves to non-compact CY threefolds, although as shown in [23] the results in the compact case are very similar. The total free energy is given by the formal power series

$$\mathcal{F}(t_a, g_s) = \sum_{g \geq 0} \mathcal{F}_g(t_a) g_s^{2g-2}, \tag{2.1}$$

where $g_s$ is the string coupling constant. It has been argued based on general arguments [18, 26] that this series is factorially divergent: for fixed $t_a$, one has

$$\mathcal{F}_g(t_a) \sim (2g)!. \tag{2.2}$$

We also recall that the free energies $\mathcal{F}_g(t_a)$ depend in addition on a choice of electric-magnetic frame, and the total free energies in different frames are related by generalized Fourier transforms [36]. It is convenient to consider arbitrary coordinates in the CY moduli space, not necessarily flat. These generic coordinates will be denoted as $z_a$, $a = 1, \cdots, n$.

The asymptotics (2.2) indicates that the theory should contain non-perturbative amplitudes, of the instanton type. In [22, 23], building on [20, 21], explicit results for these amplitudes were obtained, as well as detailed conjectures on the so-called resurgent structure of the theory [37]. The first conjecture concerns the possible singularities of the Borel transform of $\mathcal{F}(t_a, g_s)$, and it states that they occur at an integral lattice generated by the periods of the CY manifold, with the appropriate normalization. This conjecture was stated in this general form in [23], refining a previous statement [33]. To spell this out, we first recall that a choice of frame induces a choice of so-called A- and B-periods. The A-periods are given by the flat coordinates $t_a$, while the B-periods are defined by

$$\mathcal{F}_a = \frac{\partial \mathcal{F}_0}{\partial t_a}. \tag{2.3}$$

Then, instanton actions are of the form

$$\mathcal{A} = \sum_{a=1}^{n} (c_a \mathcal{F}_a + d_a t_a) + 4\pi^2 \mathrm{i} n, \tag{2.4}$$

where $n$ is an integer. With appropriate normalizations for the periods, $c_a$ and $d_a$ can be also taken to be integers. However, in this paper we will not exploit the integrality properties of the actions[1].

Our second conjecture concerns the trans-series associated to these instanton actions. If all the $c_a$ vanish, the multi-instanton amplitudes have the form obtained for the resolved conifold in [38],

$$\mathcal{F}_{\mathcal{A}}^{(\ell)} = \frac{1}{2\pi g_s}\left(\frac{\mathcal{A}}{\ell} + \frac{g_s}{\ell^2}\right)\mathrm{e}^{-\ell\mathcal{A}/g_s}, \tag{2.5}$$

where $\ell \in \mathbb{Z}_{>0}$. If the $c_a$ are not all zero, we define a modified prepotential $\mathcal{F}_0^{\mathcal{A}}$ by

$$\mathcal{A} = \sum_{a=1}^n c_a \frac{\partial \mathcal{F}_0^{\mathcal{A}}}{\partial t_a}. \tag{2.6}$$

This prepotential differs from the one in (2.3) by a second order polynomial in the $t_a$'s. Then, the one-instanton amplitude associated to the action $\mathcal{A}$ is given by

$$\mathcal{F}^{(1)} = \frac{1}{2\pi}\left(1 + g_s\sum_{a=1}^n c_a \frac{\partial \mathcal{F}}{\partial t_a}(t_b - c_b g_s, g_s)\right)\exp\left[\mathcal{F}(t_b - c_b g_s, g_s) - \mathcal{F}(t_b, g_s)\right]. \tag{2.7}$$

Here, $\mathcal{F}$ is the total free energy (2.1), in which $\mathcal{F}_0$ has been replaced by the modified prepotential $\mathcal{F}_0^{\mathcal{A}}$. In the one-modulus case $n = 1$ (the only one we will consider in this paper) we can write the action as

$$\mathcal{A} = c\frac{\partial \mathcal{F}_0}{\partial t} + dt + 4\pi^2\mathrm{i}n, \tag{2.8}$$

and we find, when $c \neq 0$,

$$\begin{aligned}
\mathcal{F}^{(1)} &= \frac{1}{2\pi}\left(1 + g_s c\frac{\partial \mathcal{F}}{\partial t}(t - cg_s, g_s)\right)\exp\left[\mathcal{F}(t - cg_s, g_s) - \mathcal{F}(t, g_s)\right] \\
&= \frac{1}{2\pi g_s}\mathrm{e}^{-\mathcal{A}/g_s}\exp\left(\frac{c^2}{2}\partial_t^2\mathcal{F}_0\right) \\
&\quad \times \left\{\mathcal{A} + g_s\left(1 - c^2\partial_t^2\mathcal{F}_0 - \mathcal{A}\left(c\partial_t\mathcal{F}_1 + \frac{c^3}{6}\partial_t^3\mathcal{F}_0\right)\right) + \mathcal{O}(g_s^2)\right\}.
\end{aligned} \tag{2.9}$$

We note that (2.7), (2.9) have to be regarded as formal trans-series, of the form

$$\mathcal{F}^{(1)} = \mathrm{e}^{-\mathcal{A}/g_s}\sum_{n\geq 0}\mathcal{F}_n^{(1)}g_s^{n-1}, \tag{2.10}$$

where the $\mathcal{F}_n^{(1)}$ can be read from (2.7), (2.9). In the one-modulus case we have, for the very first coefficients,

$$\begin{aligned}
\mathcal{F}_0^{(1)} &= \frac{\mathcal{A}}{2\pi}\,\mathrm{e}^{\frac{1}{2}c^2\mathcal{F}_0''(t)}, \\
\mathcal{F}_1^{(1)} &= -\frac{6c^2\mathcal{F}_0''(t) + \mathcal{F}_0'(t)\left(c^4\mathcal{F}_0'''(t) + 6c^2\mathcal{F}_1'(t)\right) - 6}{12\pi}\,\mathrm{e}^{\frac{1}{2}c^2\mathcal{F}_0''(t)}.
\end{aligned} \tag{2.11}$$

---

[1]Integrality issues are subtler to address in the local case, due to the noncompactness of the CY manifold.

There is a similar instanton amplitude with action $-\mathcal{A}$, and they add together to give the asymptotic behavior

$$\mathcal{F}_g(t) \sim \frac{1}{\pi}\mathcal{A}^{-2g+1}\Gamma(2g-1)\sum_{n=0}^{\infty}\frac{\mathcal{A}^n\mathcal{F}_n^{(1)}}{\Pi_{k=1}^n(2g-1-k)}, \qquad g \gg 1. \tag{2.12}$$

In practice, once the action has been identified, one considers the auxiliary sequence

$$\frac{\pi\mathcal{A}^{2g-1}}{\Gamma(2g-1)}\mathcal{F}_g(t) = \mathcal{F}_0^{(1)} + \frac{\mathcal{A}\mathcal{F}_1^{(1)}}{2g-2} + \frac{\mathcal{A}^2\mathcal{F}_2^{(1)}}{(2g-2)(2g-3)} + \mathcal{O}\big(1/g^3\big), \tag{2.13}$$

from where we can extract the instanton coefficients $\mathcal{F}_n^{(1)}$ by using standard acceleration methods, like Richardson transforms.

The expression (2.9) corresponds to the one-instanton amplitude. Explicit multi-instanton amplitudes were also determined in [22], where one can find additional information, including conjectural expressions for alien derivatives.

## 3 Large $N$ instantons in multi-cut matrix models

### 3.1 Multi-cut matrix models and their $1/N$ expansion

In this section we review some basic aspects of matrix models and their connection to topological string theory. For concreteness we will focus on Hermitian one-matrix models with a polynomial potential, although many of the results below apply to more general cases. We refer to e.g. [39] for a more detailed review. After reviewing these results, we will state our general results for large $N$ instantons in these matrix models.

The partition function of the one-matrix model is defined by the matrix integral

$$Z_N = \frac{1}{\text{vol}\,[U(N)]}\int \mathrm{d}M \, \exp\left(-\frac{1}{g_s}\text{Tr}\,V(M)\right), \tag{3.1}$$

where $V(x)$ is a polynomial potential, and $g_s$ will be identified with the topological string coupling constant. After reduction to eigenvalues we can write

$$Z_N = \frac{1}{N!}\int \prod_{i=1}^{N}\frac{\mathrm{d}\lambda_i}{2\pi}\,\Delta^2(\lambda)\,\exp\left(-\frac{1}{g_s}\sum_{i=1}^{N}V(\lambda_i)\right). \tag{3.2}$$

Here, $\Delta(\lambda)$ is the Vandermonde determinant of the eigenvalues. We want to study the model in the $1/N$ expansion, but keeping the total 't Hooft coupling

$$T = Ng_s \tag{3.3}$$

fixed. Since the potential $V(x)$ is a polynomial, it will have $s$ critical points. The most general saddle-point solution of the model, at large $N$, will be characterized by a density of eigenvalues $\rho(\lambda)$ supported on a disjoint union of $s$ intervals or cuts,

$$A_I = [x_{2I-1}, x_{2I}], \qquad I = 1, \cdots, s. \tag{3.4}$$

If the endpoints are real we will order them in such a way that $x_1 < x_2 < \cdots < x_{2s}$, but in general we can (and will) have complex cuts. When $s > 1$ this saddle-point is called an $s$-cut, or *multi-cut*

*solution*, of the Hermitian matrix model. We can define the multi-cut solution by writing the corresponding partition function as a multiple integral over eigenvalues. To do this, we note that in a $s$-cut configuration, the $N$ eigenvalues split into $s$ sets of $N_I$ eigenvalues, $I = 1, \ldots, s$, which can be written as

$$\{\lambda_{k_I}^{(I)}\}_{k_I=1,\ldots,N_I}, \qquad I = 1, \ldots, s. \tag{3.5}$$

The eigenvalues in the $I$-th set are located in the interval $A_I$, around the $I$-th extremum. We can now choose $s$ integration contours $\mathcal{C}_I$ in the complex plane, $I = 1, \ldots, s$. These contours go to infinity along directions where the integrand decays exponentially, and they have the property that each of them passes through one of the $s$ critical points (see for example [40] for a detailed argument for this). Due to this choice of integration contours, the resulting matrix integral is now convergent, and the partition function can be written as

$$Z(N_1, \ldots, N_s) = \frac{1}{N_1! \cdots N_s!} \int_{\lambda_{k_1}^{(1)} \in \mathcal{C}_1} \cdots \int_{\lambda_{k_s}^{(s)} \in \mathcal{C}_s} \prod_{i=1}^{N} \frac{\mathrm{d}\lambda_i}{2\pi} \, \Delta^2(\lambda) \exp\left(-\frac{1}{g_s} \sum_{i=1}^{N} V(\lambda_i)\right). \tag{3.6}$$

In obtaining the overall factor in (3.6) we have taken into account that there are

$$\frac{N!}{N_1! \cdots N_s!} \tag{3.7}$$

possibilities to choose the $s$ sets of $N_I$ eigenvalues. We will assume that the so-called filling fractions,

$$\epsilon_I = \frac{N_I}{N}, \qquad I = 1, 2, \ldots, s, \tag{3.8}$$

or equivalently the partial 't Hooft couplings

$$t_I = t\epsilon_I = g_s N_I \tag{3.9}$$

are fixed in the large $N$ limit. The free energy of the multi-cut matrix model at fixed filling fractions or partial 't Hooft parameters has an asymptotic $1/N$ expansion of the form

$$\mathcal{F}(N_I) = \log Z(N_I) \sim \sum_{g=0}^{\infty} \mathcal{F}_g(t_I) \, g_s^{2g-2}. \tag{3.10}$$

An important result in the theory of matrix models is that the large $N$ saddle point described by the multi-cut solution above can be encoded in a hyperelliptic curve known as the *spectral curve* of the model,

$$y^2 = \sigma(x), \tag{3.11}$$

where

$$\sigma(x) = \prod_{i=1}^{2s} (x - x_i) \tag{3.12}$$

and $x_i$ are the endpoints of the cuts. The polynomial $\sigma(x)$ is given by

$$\sigma(x) = \left(V'(x)\right)^2 + f(x), \tag{3.13}$$

where $f(x)$ is a polynomial of degree $s - 1$ that splits the $s$ double zeroes of $\left(V'(x)\right)^2$. Note in particular that the cuts appearing in the saddle-point solution correspond to $A$-periods of the spectral curve, and one has

$$t_I = \frac{1}{4\pi\mathrm{i}} \oint_{\mathfrak{a}_I} y(x)\mathrm{d}x. \tag{3.14}$$

Here, $\mathfrak{a}_I$ is a closed contour encircling the cut $A_I$. Let us note that the total 't Hooft coupling (3.3)

$$T = \sum_{I=1}^{s} t_I \tag{3.15}$$

can be evaluated by residues as a polynomial in the parameters appearing in the spectral curve. It is not really a modulus of the theory, but what is called in e.g. [42] a "mass parameter." We can then take $n = s - 1$ partial 't Hooft couplings as flat coordinates parametrizing the moduli space of the theory.

The planar free energy $\mathcal{F}_0(t_I)$ can be computed as follows. Let us consider the cuts $B_I$, $I = 1, \cdots, s - 1$, going from the end of the $A_I$ cut to the beginning of the $A_{I+1}$ cut. Then, the dual periods

$$t_{D,I} = \int_{B_I} y(x)\mathrm{d}x\,, \qquad I = 1, \ldots, s - 1 \tag{3.16}$$

are related to the planar free energy as

$$t_{D,I} = \frac{\partial \mathcal{F}_0}{\partial t_I} - \frac{\partial \mathcal{F}_0}{\partial t_{I+1}}. \tag{3.17}$$

The higher genus free energies $\mathcal{F}_g(t_I)$ appearing in the $1/N$ expansion (3.10) can also be obtained in various ways. Perhaps the most powerful and deeper approach to this problem is topological recursion [13, 43], although we will not need this method in this paper.

The series (3.10) has the form of an asymptotic expansion in topological string theory, and indeed it was argued in [2] that it can be regarded as the free energy of topological string theory on a non-compact CY of the form

$$uv = y^2 - \sigma(x). \tag{3.18}$$

The connection to topological strings suggests that the $\mathcal{F}_g(t_I)$ can also be computed by using the HAE of [8]. This was first used in [11], and then proved in full generality in [12] as a consequence of the topological recursion of [13]. In order to actually compute the $\mathcal{F}_g$s of multi-cut matrix models, the HAE turn out to be more efficient than topological recursion, and this is the method we will use in this paper, as we explain below.

The moduli space of CY threefolds has singular loci which lead to a singular behavior in the genus $g$ free energies. In the case of the CY geometry associated to matrix models, these are the loci where the discriminant $\Delta$ of the spectral curve (3.11) vanishes, and at least two of the roots $x_i$, $i = 1, \cdots, 2s$ come together. The loci with smaller codimension correspond to the case in which one 't Hooft coupling $t_J$ vanishes, and the corresponding A-cycle shrinks to zero size, or to the case in which one dual period $t_{D,J}$ vanishes, and the dual cut $B_J$ shrinks. The effect of a vanishing A-period in the genus $g$ free energies is well-known, and leads to a singular behavior

$$\mathcal{F}_g \sim \frac{B_{2g}}{2g(2g-2)} t_J^{2-2g} + \mathcal{O}(1), \tag{3.19}$$

where $B_{2g}$ are Bernoulli numbers. This is the famous gap condition for the free energies, which was much exploited in [11]. In general CY manifolds, the gap condition is a deep statement on the universal behavior at the conifold point [44]. In the case of matrix models, the gap condition follows from conventional perturbation theory and the structure of the Gaussian matrix model, see e.g. [45]. When there is a vanishing B-cycle, one has to perform a symplectic transformation to a frame in which the dual vanishing cycle $t_{D,J}$ becomes a flat coordinate. One then has the

same behavior (3.19) for the dual free energies. This was exploited in [30] to obtain free energies at large genus from the HAE in certain cases, as we will review below.

The series in the r.h.s. of (3.10) is factorially divergent, and one can ask what is its resurgent structure, in the sense explained in [22, 37]. This means that we would like to know what are the possible actions characterizing multi-instantons, and what are the corresponding amplitudes. Since the $1/N$ expansion (3.10) is a particular case of a topological string free energy, it follows that the results of [20–23] must describe the resurgent structure of the $1/N$ expansion in generic multi-cut matrix models. A basis for the periods of the underlying CY manifold can be taken to be a subset of $s - 1$ partial 't Hooft couplings, $t_a$, $a = 1, \cdots, s - 1$, and the dual periods $t_{D,a}$, $a = 1, \cdots, s - 1$. The general action characterizing an instanton sector will be given by

$$\mathcal{A} = \sum_{a=1}^{s-1} (c_a t_a + d_a t_{D,a}) + 4\pi^2 \mathrm{i} \gamma, \tag{3.20}$$

and the corresponding instanton amplitudes are given by the general expression (2.7). This is our proposal for large $N$ instantons in generic matrix models. As we mentioned in the introduction, the basis for this proposal is simply that the free energies appearing in the $1/N$ expansion of the matrix model satisfy the HAE. The instanton amplitudes obtained in [22, 23] are trans-series solutions to the HAE, and therefore they should apply as well to the case of matrix models. There is an additional ingredient in the derivation of [22, 23], namely boundary conditions fixing the holomorphic ambiguity in the trans-series. These boundary conditions lead to the expression (2.5), and they are fixed, as first explained in [20, 21], by the behavior of the free energies at singular loci. In the case of matrix models, this behavior is given by (3.19), which is the conifold behavior of topological strings, and therefore it leads to the same boundary conditions and to the behavior (2.5). In the remaining of this section, we will test our proposal in the simplest multi-cut matrix model, namely the cubic, two-cut matrix model.

## 3.2 Testing the large $N$ instantons

### 3.2.1 The cubic matrix model and its $1/N$ expansion

The simplest two-cut matrix model has a cubic potential. The one-cut case of the cubic potential was already considered in [46], and the two-cut case has been studied intensively. A non-exhaustive list of references includes [11, 41, 47–49]. We will closely follow [30].

Without lack of generality, we can take the potential of the cubic matrix model to be

$$V(x) = \frac{x^3}{3} - x, \tag{3.21}$$

which is represented in Fig. 1. Therefore, the most general two-cut phase of the cubic matrix model is described by the spectral curve (3.11), where $\sigma(x)$ is given by (3.13) and $f(x)$ has degree one. We write this curve as

$$y^2 = (x^2 - 1)^2 + \alpha x - z, \tag{3.22}$$

where $\alpha$ and $z$ are parameters. There are two cuts $[x_1, x_2]$, $[x_3, x_4]$ and two partial 't Hooft couplings, which we will denote as[2]

$$t_2 = \frac{1}{2\pi \mathrm{i}} \int_{x_1}^{x_2} y(x) \mathrm{d}x, \qquad t_1 = \frac{1}{2\pi \mathrm{i}} \int_{x_3}^{x_4} y(x) \mathrm{d}x. \tag{3.23}$$

---

[2]For convenience we have exchanged their labels w.r.t. what we have in (3.14).

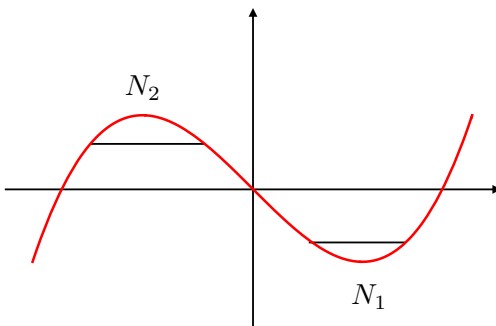

**Figure 1**: The potential (3.21) of the cubic matrix model, as a function of $x$. In the two-cut configuration, $N_1$ eigenvalues sit near the stable critical point at $x = 1$, and $N_2$ eigenvalues sit at the unstable critical point at $x = -1$.

The dual period (3.16) is given by

$$t_D = \int_{x_2}^{x_3} y(x)\mathrm{d}x. \tag{3.24}$$

It turns out that $\alpha$ and $z$ have a very different geometric meaning. $\alpha$ is related to the total 't Hooft parameter, and one can easily show by a contour deformation argument that:

$$T = t_1 + t_2 = -\frac{\alpha}{4}. \tag{3.25}$$

As we mentioned before, $\alpha$ is a "mass parameter," while $z$ is a true modulus. We will denote $t = t_1$. Sometimes we we will only indicate the dependence of the free energies on the flat coordinate corresponding to the true modulus, and we will write $\mathcal{F}_g(t)$.

The large $N$ expansion of the cubic matrix model in the general two-cut phase has been considered in many papers. The genus zero free energy was studied in e.g. [47]. The genus one free energy was first obtained for generic two-cut matrix models in [50] and further studied e.g. in [41]. It is given by the formula

$$\mathcal{F}_1 = -\frac{1}{2}\log\left(\frac{\partial t}{\partial z}\right) - \frac{1}{12}\log\Delta, \tag{3.26}$$

where $\Delta$ is the discriminant of the spectral curve. In our case it is easily computed to be

$$\Delta = 256z^2(1-z) + 32\alpha^2(9z-8) - 27\alpha^4. \tag{3.27}$$

In addition, we have

$$\frac{\partial t}{\partial z} = \frac{2}{\sqrt{(x_1 - x_3)(x_2 - x_4)}}K(k), \tag{3.28}$$

where $K(k)$ is the elliptic function of the first kind with modulus

$$k^2 = \frac{(x_1 - x_2)(x_3 - x_4)}{(x_1 - x_3)(x_2 - x_4)}. \tag{3.29}$$

The higher genus corrections were obtained with the HAE of [8]. In [11] explicit results were presented for $\mathcal{F}_2$, while in [30] results were obtained up to $g = 4$. Both references regarded the

geometry as a two-parameter problem. In order to explore the asymptotics of the $1/N$ expansion we need more terms in the genus expansion than what was obtained in [11, 30]. To do this we will regard the geometry as a one-modulus problem with a mass parameter $\alpha$. This makes it possible to calculate the genus expansion up to $g = 18$, which is enough to clearly see the asymptotics in various regions. Before presenting our results, let us quickly review the formalism of the HAE, in the one-modulus case, following [22].

In the HAE, the genus $g$ free energies are regarded as functions of a complex coordinate $z$, which parametrizes the moduli space, and of a propagator function $S$, which is a *non-holomorphic* function of $z$. They can also depend on global parameters, like $\alpha$ in our case, but we will not always indicate this dependence explicitly. The non-holomorphic free energies will then be denoted by $F_g(S, z)$, $g \geq 2$, as opposed to their holomorphic counterparts $\mathcal{F}_g$. The moduli space can also be parametrized by a so-called flat coordinate, denoted by $t$. It is given by an appropriate period of the CY and related to $z$ by a mirror map $t(z)$. In the case of the cubic matrix model, we will take as complex parameter the $z$ entering in the spectral curve (3.22), and as we mentioned above, $t$ is just the partial 't Hooft parameter $t_1$.

The propagator $S$ plays a central rôle in the theory of HAE. It is related to the non-holomorphic genus one free energy through the equation

$$\partial_z F_1 = \frac{1}{2} C_z S + \text{holomorphic}. \tag{3.30}$$

Here, $C_z$ denotes the so-called Yukawa coupling in the $z$ coordinate, which is defined by

$$\partial_t^3 \mathcal{F}_0 = C_t = \left(\frac{\mathrm{d}z}{\mathrm{d}t}\right)^3 C_z. \tag{3.31}$$

The holomorphic function in the r.h.s. of (3.30) can be regarded as a choice of "gauge" for the propagator. The holomorphic free energies $F_g(S, z)$ is obtained by taking the so-called holomorphic limit of the propagator, which will be denoted by $\mathcal{S}$. It is a holomorphic function of $z$ and the parameters. We then have

$$\mathcal{F}_g(t) = F_g\left(S = \mathcal{S}(z), z\right), \tag{3.32}$$

after one expresses $z$ as a function of $t$.

As we explained above, there are various choices of "frame" for the holomorphic free energies $\mathcal{F}_g$, which are characterized by different choices of flat coordinates $t$. Correspondingly, the propagator $S$ has different holomorphic limits depending on the frame one chooses. A convenient aspect of the HAE is that the holomorphic free energies in a given frame can be obtained from the *same* function $F_g(S, z)$ by choosing different holomorphic limits for $S$ and different inverse mirror maps $t(z)$. Of course, in the case of matrix models there is a preferred frame corresponding to the large $N$ expansion of the matrix integral, but there are other choices one can consider. As we have mentioned, there are "dual" frames in which the flat coordinates include dual periods like (3.24).

There is a very useful formula which expresses the holomorphic limit of $S$ in terms of the mirror map $t(z)$ for the corresponding flat coordinate:

$$\mathcal{S} = -\frac{1}{C_z} \frac{\mathrm{d}^2 t}{\mathrm{d}z^2} \frac{\mathrm{d}z}{\mathrm{d}t} - \mathfrak{s}(z). \tag{3.33}$$

Here, $\mathfrak{s}(z)$ is a holomorphic function of $z$ which is independent of the frame, and encodes the choice of gauge for the propagator that we mentioned above. The propagator satisfies various

important properties. The first one, which follows from the so-called special geometry of the CY moduli space, is that its derivative w.r.t. $z$ can be written as a quadratic polynomial in $S$:

$$\partial_z S = S^{(2)}, \qquad S^{(2)} = C_z \left( S^2 + 2\mathfrak{s}(z)S + \mathfrak{f}(z) \right), \tag{3.34}$$

where $\mathfrak{f}(z)$ is again a universal, holomorphic function independent of the frame.

Let us now write down the HAE of BCOV, in the case at hand. These equations determine the dependence of $F_g(S, z)$ on the propagator, once the lower order functions $F_{g'}(S, z)$, $g' < g$, are known. They read,

$$\frac{\partial F_g}{\partial S} = \frac{1}{2} \left( D_z^2 F_{g-1} + \sum_{m=1}^{g-1} D_z F_m D_z F_{g-m} \right), \qquad g \geq 2. \tag{3.35}$$

Here, $D_z$ is the covariant derivative w.r.t. the metric on the Kähler moduli space. Its Christoffel symbol is related to the propagator through

$$\Gamma_{zz}^z = -C_z \left( S + \mathfrak{s}(z) \right). \tag{3.36}$$

In the case of the two-cut matrix model, a clever choice of the propagator simplifies the tasks enormously. Such a choice is equivalent to a choice of function $\mathfrak{s}$ in (3.33), which determines uniquely the function $\mathfrak{f}$ in (3.34). It turns out that the values

$$\mathfrak{s}(z, \alpha) = -\frac{6 \left( -16\alpha^2 + 16z^2 + 3\alpha^2 z \right)}{16z - 9\alpha^2},$$
$$\mathfrak{f}(z, \alpha) = \frac{36 \left( 3\alpha^4 + 16z^3 - \alpha^2 z^2 - 16\alpha^2 z \right)}{16z - 9\alpha^2}, \tag{3.37}$$

are very convenient, and this is what we used in our calculations. In addition, the Yukawa coupling reads

$$C_z = \frac{16z - 9\alpha^2}{2\Delta}. \tag{3.38}$$

The HAE determines the $F_g(S, z)$ as a polynomial in the propagator, but one has an integration constant $f_g(z)$ at every genus $g \geq 2$ which is usually called the *holomorphic ambiguity*. Determining $f_g(z)$ is a subtle task. One usually needs an ansatz for it, as a rational function on the moduli space with possible singularities at special points. In the case of the two-cut matrix model, we expect the holomorphic ambiguity to be of the form

$$f_g(z) = \frac{1}{\Delta^{2g-2}} p_g(z, \alpha^2), \tag{3.39}$$

where $p_g(z, \alpha^2)$ is a polynomial. We will assign the degrees 2 and 3 to $z$ and $\alpha^2$. Then, $\Delta$ has degree 6, and the denominator appearing in (3.39) has degree $12(g-1)$. We will assume that the numerator is a polynomial of the same degree, i.e.

$$p_g(z, \alpha^2) = \sum_{i,j \geq 0} a_{ij} z^i \alpha^{2j}, \qquad 2i + 3j \leq 12(g-1). \tag{3.40}$$

This will be our ansatz for the ambiguity. We now consider the simultaneous limit $t_{1,2} \to 0$, where due to (3.19) one has the gap condition

$$\mathcal{F}_g(t_1, t_2) \sim \frac{B_{2g}}{2g(2g-2)} \left( \frac{1}{t_1^{2g-2}} + \frac{1}{t_2^{2g-2}} \right) + \mathcal{O}(t_1, t_2). \tag{3.41}$$

It turns out that this behaviour fixes the ambiguity completely, as noted in [30]. In practice, and in order to implement the gap condition (3.41), it is not convenient to use $z$ and $\alpha$, since the expressions of $t_{1,2}$ in terms of these parameters are complicated. There is a convenient reparametrization, first introduced in [47] and reviewed in the Appendix, which uses two complex parameters $z_{1,2}$. The locus $t_1 = t_2 = 0$ corresponds to $z_1 = z_2 = 0$. By expanding everything in power series in these two new parameters around $z_1 = z_2 = 0$, it is possible to fix systematically the holomorphic ambiguities. One finds for example, for $g = 2$, and with the above choice of the propagator,

$$
\begin{aligned}
p_2(z, \alpha^2) = & -\frac{2322\alpha^6}{5} - \frac{32256\alpha^4}{5} - \frac{524288\alpha^2}{15} + \frac{27200z^5}{3} - 1704\alpha^2 z^4 - 50176z^4 \\
& + \frac{135\alpha^4 z^3}{4} + \frac{115008\alpha^2 z^3}{5} + \frac{229376z^3}{3} - 1728\alpha^4 z^2 - \frac{1091072\alpha^2 z^2}{15} \\
& - \frac{524288z^2}{15} + \frac{42816\alpha^4 z}{5} + \frac{425984\alpha^2 z}{5}.
\end{aligned}
\tag{3.42}
$$

The generic two-cut cubic matrix model is relatively involved, and this is the reason that we can only obtain the genus expansion up to relatively low genus. It is therefore natural to search for a simpler case which can be still regarded as a *bona fide* two-cut example. It turns out that the theory simplifies enormously when $\alpha = 0$. In this slice, the spectral curve becomes

$$
y^2 = (x^2 - 1)^2 - z,
\tag{3.43}
$$

which as noted in [48], it is nothing but the Seiberg–Witten curve for pure $\mathcal{N} = 2$ super Yang–Mills theory [51]. It describes the cubic matrix model in which the partial 't Hooft parameters satisfy

$$
t_1 = -t_2.
\tag{3.44}
$$

There are various manifestations of the underlying simplicity of the theory at $\alpha = 0$. For example, the period $t = t_1$ and its dual $t_D$ can be written explicitly in terms of elliptic integrals of the first and second kind as

$$
\begin{aligned}
t &= \frac{\sqrt{1 + \sqrt{z}}}{3\pi} \left[ E\left(\frac{2\sqrt{z}}{1 + \sqrt{z}}\right) + (\sqrt{z} - 1)K\left(\frac{2\sqrt{z}}{1 + \sqrt{z}}\right) \right], \\
t_D &= \frac{1}{2\pi i} \frac{4\sqrt{1 + \sqrt{z}}}{3} \left[ E\left(\frac{1 - \sqrt{z}}{\sqrt{z} + 1}\right) - \sqrt{z}K\left(\frac{1 - \sqrt{z}}{\sqrt{z} + 1}\right) \right].
\end{aligned}
\tag{3.45}
$$

In addition, and most important to us, when $\alpha = 0$ it is possible to solve the HAE to large genus. This was already noted in [30]. As usual the key issue is to fix the holomorphic ambiguity, and in this case this is done as follows. When $\alpha = 0$ there are two singular points in the moduli space parametrized by $z$. The point $z = 0$ corresponds to $t = 0$, and we can use the gap condition (3.41), which on this slice reads

$$
\mathcal{F}_g(t) \sim \frac{B_{2g}}{g(2g - 2)} \frac{1}{t^{2g-2}} + \mathcal{O}(t).
\tag{3.46}
$$

The other singular point occurs at $z = 1$, where the dual period vanishes: $t_D = 0$. Let us then consider the frame associated to the dual period $t_D$, and let us denote by $\mathcal{F}_g^D(t_D)$ the

corresponding dual free energies. Then, near $z = 1$ the dual free energies have a singular behavior, which is described by the dual gap condition [30]

$$\mathcal{F}_g^D(t_D) \sim \frac{B_{2g}}{2g(2g-2)} \frac{1}{t_D^{2g-2}} + \mathcal{O}(1). \tag{3.47}$$

By using these two gap conditions, one can compute the $\mathcal{F}_g(t)$ up to very high genus, say $g \sim 100$. This is very useful to do precision tests of our results for large $N$ instantons.

### 3.2.2 Asymptotics and large $N$ instantons

We will now test that the topological string instanton amplitude given in (2.7), (2.9) provides the appropriate large $N$ instanton amplitude, in the case of the two-cut matrix model at generic points in moduli space.

We first consider the slice where $\alpha = 0$, since in this case we can compute many terms in the $1/N$ expansion. As noted in [20, 21], the gap behavior (3.46) implies that there is a Borel singularity with action given by

$$\mathcal{A} = 2\pi \mathrm{i} t. \tag{3.48}$$

This leads to "trivial" instanton amplitudes of the form (2.5). The effect of this singularity can be completely subtracted by simply considering

$$\mathcal{G}_g(t) \equiv \mathcal{F}_g(t) - \frac{B_{2g}}{g(2g-2)} \frac{1}{t^{2g-2}}. \tag{3.49}$$

In order to look for Borel singularities of $\mathcal{F}_g(t)$ other than (3.48), one simply considers the Borel singularities associated to the series of subtracted free energies $\mathcal{G}_g(t)$. An additional Borel singularity is obtained by considering the behavior of the dual free energy (3.47). It occurs at

$$\mathcal{A}_D = 2\pi \mathrm{i} t_D. \tag{3.50}$$

This leads to a non-trivial instanton amplitude, since

$$\mathcal{A}_D = \partial_t \mathcal{F}_0, \tag{3.51}$$

and we have $c = 1$ in (2.6). The amplitude is given by the general expression (2.9), and it leads to a prediction for the large genus asymptotics of the $\mathcal{F}_g$s which can be tested with high precision. In practice, as in [27], we construct auxiliary sequences like (2.13) which asymptote to the values $\mathcal{F}_n^{(1)}$, for $n = 0, 1, \cdots$. After using standard acceleration methods we obtain numerical estimates of the asymptotic values, which can then be compared with the instanton predictions in e.g. (2.11). In Fig. 2 we make such a comparison, finding excellent agreement. The red line is the theoretical prediction for $\mathcal{F}_n^{(1)}$, $n = 0, 1, 2$, as a function of the modulus $z$, while the black dots are numerical estimates obtained from the perturbative series up to $g = 135$. The error bars in the numerical results are estimated from the difference between two successive Richardson transforms. To find the best asymptotic estimate for $\mathcal{F}_n^{(1)}$, we perform a number of Richardson transforms so that this error is minimized. We note that, for points sufficiently close to $z = 1$, the relative error of our numerical asymptotic estimates is as small as $10^{-28}$, but it increases as we approach the point $z = 0$. This is related to the fact that, near $z = 0$, the action $\mathcal{A}_D$ becomes larger.

Although the slice $\alpha = 0$ is a generic submanifold of the moduli space of the two-cut matrix model, it is important to make sure that the topological string instanton amplitudes describe

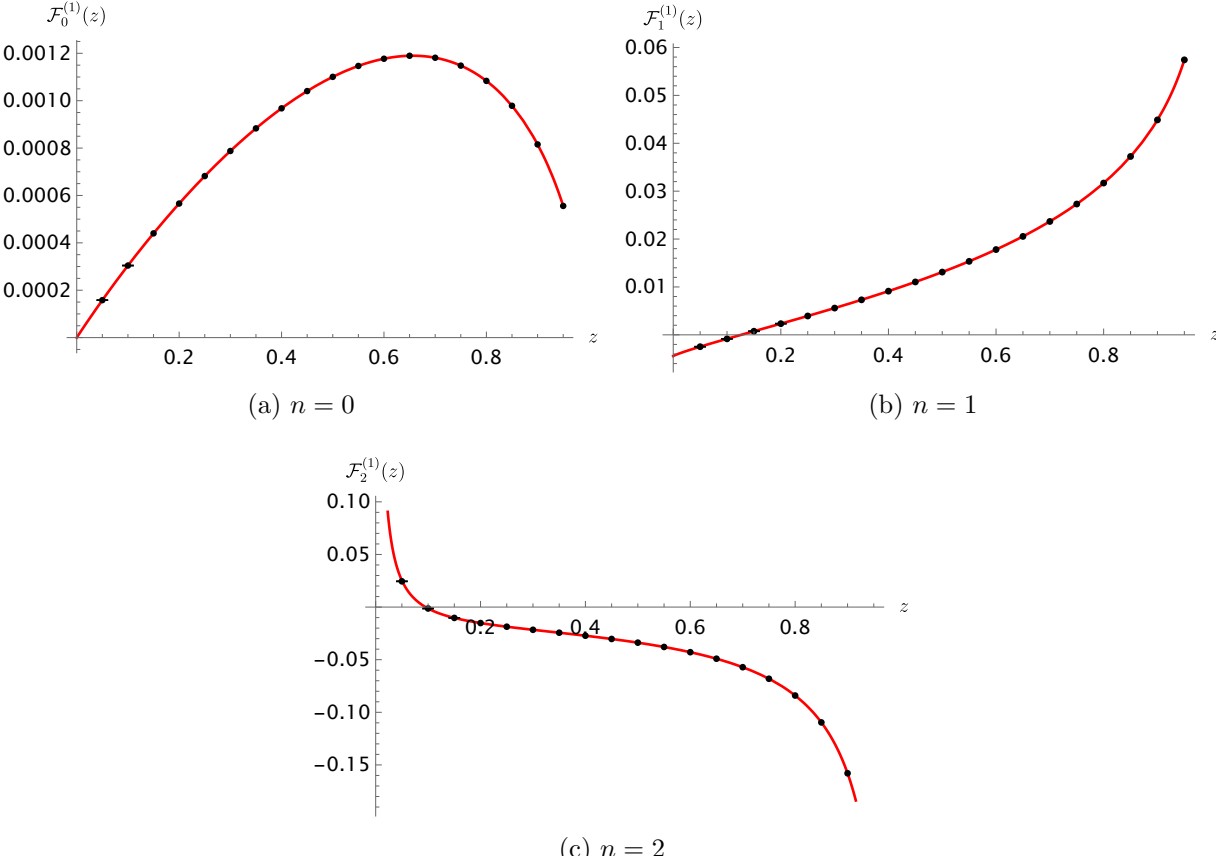

(a) $n = 0$

(b) $n = 1$

(c) $n = 2$

**Figure 2**: Coefficients $\mathcal{F}_n^{(1)}$, for $n = 0, 1, 2$, as a function of $z$, for the cubic matrix model at the slice $\alpha = 0$. The red line is the analytic result predicted from (2.9). The black dots are the numerical approximations extracted from the large order behaviour of the sequence $\mathcal{F}_g$, $g = 2, \cdots, 135$.

the appropriate large $N$ instantons for arbitrary values of $\alpha$. Fortunately, we have computed the general $\mathcal{F}_g(t_1, t_2)$ up to $g = 18$, and this is enough to check quantitatively that its large genus asymptotics is still controlled by (2.9). We note that the derivatives w.r.t. $t$ in (2.9) are computed at constant $\alpha$, therefore $t_2$ depends on $t_1$, as follows from (3.25), and

$$\partial_t \mathcal{F}(t_1, t_2) \equiv \partial_t \mathcal{F}(t, -t - \alpha/4). \tag{3.52}$$

Due to the gap condition (3.41), there are singularities at $\mathcal{A}_{1,2} = 2\pi i t_{1,2}$. We can remove their effect by considering the subtracted quantity

$$\mathcal{G}_g(t_1, t_2) = \mathcal{F}_g(t_1, t_2) - \frac{B_{2g}}{2g(2g-2)} \left( \frac{1}{t_1^{2g-2}} + \frac{1}{t_2^{2g-2}} \right). \tag{3.53}$$

There will be a Borel singularity at the dual action (3.50), as in the case of $\alpha = 0$ (although $t_D$ will be given by a more complicated formula than the one in (3.45)). When comparing the asymptotics with the instanton predictions there are two cases to consider. The simplest one is when the action $\mathcal{A}$ is real. We can then extract numerical estimates for the coefficients $\mathcal{F}_n^{(1)}$, for

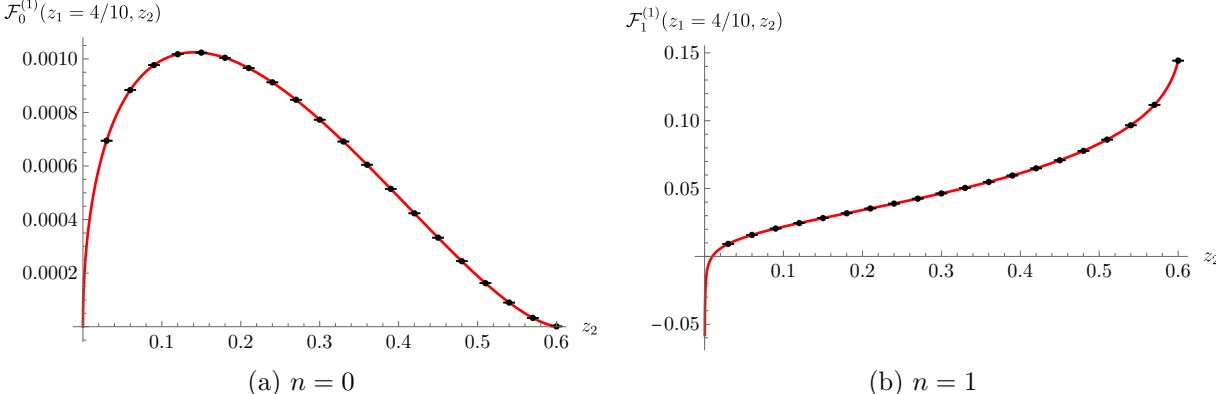

(a) $n = 0$                              (b) $n = 1$

**Figure 3**: Coefficients $\mathcal{F}_n^{(1)}$ for $n = 0, 1$ in the cubic matrix model, as a function of $z_2$, for fixed $z_1 = 2/5$. The red line is the analytic result predicted from (2.9). The black dots are the numerical approximations extracted from the large order behaviour of the sequence $\mathcal{F}_g$, for $g = 2, \cdots, 18$.

different values of the moduli, and compare them to the prediction. It is useful to parametrize the moduli space with the coordinates $z_{1,2}$ introduced in the Appendix. For convenience, we fix the value of $z_1$ and we vary the value of $z_2$. In Fig. 3 we plot $\mathcal{F}_{0,1}^{(1)}$ as a function of $z_2$, and we indicate the numerical estimates obtained from the asymptotics. $z_1$ is taken to be $2/5$, while the numerical estimates are made for values of $z_2$ of the form

$$z_2 = \frac{3i}{100}, \qquad i = 1, \cdots, 20. \tag{3.54}$$

We note that these values of the parameters lead to $t_1 > 0$, $t_2 < 0$. As we can see, the agreement between the prediction and the empirical data is excellent. With our data for the $\mathcal{F}_g$s, $0 \leq g \leq 18$, we obtain estimates for $\mathcal{F}_n^{(1)}$, $n = 0, 1$ with a relative error not worse than $10^{-6}$.

The other case to consider is when the dual action is complex. This happens for example when $z_1 > 0$ and $z_2 < 0$ and both are sufficiently small. It corresponds to the case in which $t_{1,2} > 0$. As it is well-known, when the action is complex, both the action and its complex conjugate $\overline{\mathcal{A}}$ contribute to the asymptotics, which is oscillatory. Let us write

$$\mathcal{A} = |\mathcal{A}| e^{i\theta_\mathcal{A}}, \qquad \mathcal{F}_n^{(1)} = \left|\mathcal{F}_n^{(1)}\right| e^{i\theta_{\mathcal{F}_n^{(1)}}}. \tag{3.55}$$

When the asymptotics is oscillatory, it is more difficult to use acceleration methods. To perform our tests, we consider the normalized free energies:

$$\widehat{\mathcal{G}}_g(t_1, t_2) = \frac{\pi \mathcal{G}_g(t_1, t_2) |\mathcal{A}|^{2g-1}}{\left|\mathcal{F}_0^{(1)}(t_1, t_2)\right| \Gamma(2g-1)}. \tag{3.56}$$

They have the asymptotic behavior

$$\begin{aligned}
\widehat{\mathcal{G}}_g(t_1, t_2) &\sim \sum_{n=0}^{\infty} \frac{|\mathcal{A}|^n \left|\mathcal{F}_n^{(1)}(t_1, t_2)\right|}{\left|\mathcal{F}_0^{(1)}(t_1, t_2)\right| \Pi_{k=1}^n (2g + b - k)} 2\cos\left(-(2g - 1 - n)\theta_\mathcal{A} + \theta_{\mathcal{F}_n^{(1)}}\right) \\
&\sim 2\cos\left(-(2g - 1)\theta_\mathcal{A} + \theta_{\mathcal{F}_0^{(1)}}\right) + \mathcal{O}(1/g).
\end{aligned} \tag{3.57}$$

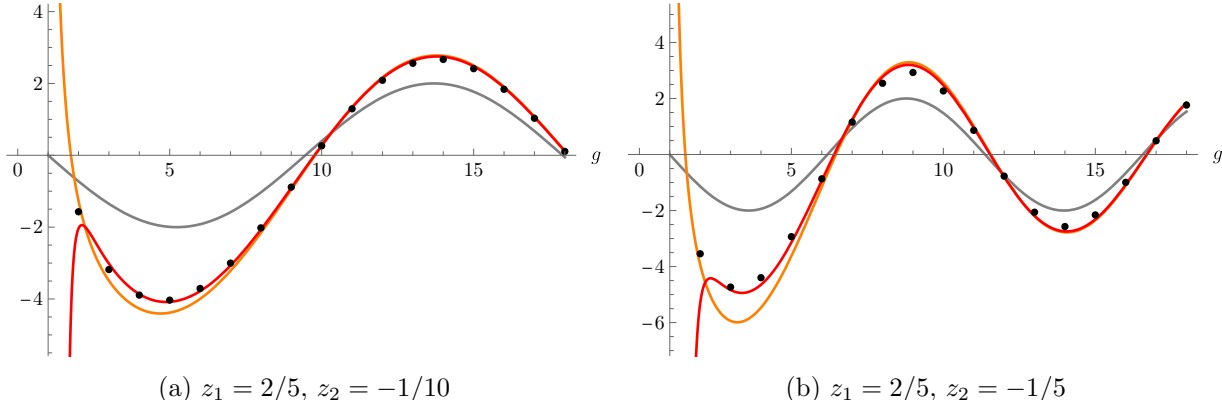

(a) $z_1 = 2/5$, $z_2 = -1/10$          (b) $z_1 = 2/5$, $z_2 = -1/5$

**Figure 4**: Normalized free energies $\widehat{\mathcal{G}}_g(t)$ for the cubic matrix model (black dots) as compared to the prediction (3.57) for the asymptotics (lines). In grey, we include the leading term; in orange, the subleading term; and, in red, we include the subsubleading term.

so we simply compare the prediction obtained by truncating the r.h.s. of (3.57), to the sequence in the l.h.s. This is done in Fig. 4 for two points in the moduli space, which we label by the parameters $z_{1,2}$ introduced in the Appendix. We see that, as we add more terms in the sum of the r.h.s. of (3.57), we find better approximations for $\widehat{\mathcal{G}}_g(t_1, t_2)$. This is specially clear for low values of $g$, in which the corrections lead to a substantial improvement.

In this paper we have focused on one-instanton amplitudes, but there are Borel singularities at e.g. integer multiples $\ell \mathcal{A}_D$, with $\ell \in \mathbb{Z}_{>0}$, leading to $\ell$-instanton amplitudes. Explicit expressions for these amplitudes can be found in [22, 23]. In the case of the cubic matrix model with $\alpha = 0$, we have verified the expression for the two-instanton amplitude of [22, 23] by calculating numerically the Stokes discontinuity of the free energies.

### 3.2.3    On the one-cut limit

When there are no eigenvalues in the unstable critical point of the cubic matrix model, $t_2 = 0$ and one recovers the one-cut matrix model studied in the seminal paper [46]. The one-cut free energies are obtained as

$$\mathcal{F}_g(t) = \lim_{t_2 \to 0} \left\{ \mathcal{F}_g(t, t_2) - \frac{B_{2g}}{2g(2g-2)} \left( \frac{1}{t^{2g-2}} + \frac{1}{t_2^{2g-2}} \right) \right\}, \qquad g \geq 2, \qquad (3.58)$$

and a similar formula holds for $g = 0, 1$, where one has to subtract logarithmic divergences. The large genus asymptotics of the one-cut free energies was studied in [27], where one-instanton amplitudes were studied by using eigenvalue tunneling. It is therefore natural to try to obtain the one-instanton amplitudes of [27] as a limit of the generic multi-cut instanton amplitude (2.7) studied in this paper. However, one should note that the instanton results of [27] are qualitatively different from the ones found here for the generic two-cut case. For example, the large genus asymptotics obtained in [27] in the one-cut case involves a factorial growth of the form $\Gamma(2g-5/2)$, while in the two-cut case we find the growth $\Gamma(2g-1)$.

What one finds is that the one-cut limit of the generic two-cut instanton amplitude is singular. This is because it involves derivatives of the free energies $\mathcal{F}_g(t_1, t_2)$, which are singular due precisely to the polar terms that are being subtracted in (3.58). In addition, we have evidence

that the large genus asymptotics of the free energies $\mathcal{F}_g(t_1, t_2)$ changes *discontinuously* as we take the one-cut limit. Our results seem to indicate that, for any $t_2 \neq 0$, no matter how small, the asymptotics is controlled by (2.9), and it is only when we set $t_2 = 0$ and we subtract the polar part as in (3.58) that the asymptotics is governed by the one-instanton amplitude of [27]. In this sense, it does not seem possible (or at least, straightforward) to interpolate smoothly between the generic two-cut case studied in this paper and the one-cut case of [27].

## 4 Large $N$ instantons in ABJM theory

### 4.1 The ABJM matrix model and its $1/N$ expansion

ABJM theory [3] is an important example of a large $N$ duality, relating string/M-theory on an AdS$_4$ compactification to a superconformal Chern–Simons–matter theory. It turns out that the free energy on the three-sphere of the field theory realization can be computed in terms of a matrix model, by using localization [4] (see [52] and the collection of articles [53] for an extensive discussion). It was found in [5, 6] that the resulting matrix model is equivalent to topological string on a toric geometry, called the local $\mathbb{F}_0$ geometry, and this allows to determine its $1/N$ expansion at all orders by using the HAE. Non-perturbative aspects of the matrix model of ABJM theory were addressed in [33], which studied in particular the large order behavior of the $1/N$ expansion. However, a precise determination of the large $N$ instantons of this theory was not available in [33]. We will now show that the topological string instantons of [22] describe the large $N$ instantons of the ABJM matrix models. It was conjectured in [33] that some of the large $N$ instantons of the ABJM matrix model correspond to D2-branes in the large $N$ dual string background. Therefore, the instanton amplitude obtained in [22] should provide a precise prediction for the D2-brane amplitude, at all orders in the string coupling constant.

Let us first summarize some relevant facts on the ABJM matrix model and its $1/N$ expansion, and refer to [5, 6, 33, 52–54] for more details. The partition function is given by the matrix integral

$$Z(N, g_s) = \frac{1}{(N!)^2} \int \prod_{i=1}^{N} \frac{\mathrm{d}\mu_i \mathrm{d}\nu_i}{(2\pi)^2} \frac{\prod_{i<j} \left(2\sinh\left(\frac{\mu_i - \mu_j}{2}\right)\right)^2 \left(2\sinh\left(\frac{\nu_i - \nu_j}{2}\right)\right)^2}{\prod_{i,j} \left(2\cosh\left(\frac{\mu_i - \nu_j}{2}\right)\right)^2} \mathrm{e}^{-\frac{1}{2g_s}\sum_i \left(\mu_i^2 - \nu_j^2\right)}. \quad (4.1)$$

The string coupling constant $g_s$ is related to the Chern–Simons coupling $k$ by

$$g_s = \frac{2\pi \mathrm{i}}{k}, \quad (4.2)$$

and the 't Hooft coupling is usually taken to be

$$\lambda = \frac{N}{k}. \quad (4.3)$$

The matrix model free energy has a $1/N$ expansion of the form

$$\mathcal{F}(\lambda, g_s) = \sum_{g \geq 0} \mathcal{F}_g(\lambda) g_s^{2g-2}. \quad (4.4)$$

It was found in [5] that this expansion corresponds to the topological string on the so-called local $\mathbb{F}_0$ geometry, and in a special frame called the orbifold frame. The moduli space of this geometry is parametrized by a complex coordinate that we will denote again by $z$ (the local $\mathbb{F}_0$ geometry

also has a "mass parameter" $m$, but in order to obtain the ABJM theory we have to set it to $m = 1$; more general values of $m$ correspond to a generalization of ABJM theory called ABJ theory [55], which we will not consider in this paper).

The geometric ingredients which are needed to obtain the $1/N$ expansion of the ABJM matrix model from the HAE are the same ones introduced in the previous section on the cubic matrix model. The discriminant and Yukawa coupling are given by

$$\Delta = 1 - 16z, \qquad C_z = \frac{1}{4z^3\Delta}. \tag{4.5}$$

The orbifold coordinate, appropriate for the ABJM matrix model, is given by

$$t_o = \frac{1}{4\sqrt{z}} \, {}_3F_2\left(\frac{1}{2}, \frac{1}{2}, \frac{1}{2}; 1, \frac{3}{2} \, \middle| \, \frac{1}{16z}\right), \tag{4.6}$$

and it gives the 't Hooft parameter as a function of the modulus $z$,

$$t_o = Ng_s = \frac{\lambda}{2\pi\mathrm{i}}. \tag{4.7}$$

Together with (3.31), the data above determine the large $N$ free energy $\mathcal{F}_0(\lambda)$ or prepotential (up to a quadratic polynomial in $\lambda$). They also determine the genus one free energy through the expression

$$\mathcal{F}_1(\lambda) = -\frac{1}{2}\log\left(-\frac{\mathrm{d}t_o}{\mathrm{d}z}\right) - \frac{1}{12}\log\left(z^7\Delta\right). \tag{4.8}$$

To obtain the higher genus free energies we have to solve the HAE. A convenient choice of propagator is specified by the functions

$$\mathfrak{s}(z) = -\frac{2}{3}z^2\left(128z - 7\right),$$
$$\mathfrak{f}(z) = \frac{4}{9}z^4\left(256z^2 - 16z + 1\right). \tag{4.9}$$

The holomorphic ambiguity is of the form

$$f_g(z) = \frac{\sum_{n=0}^{3g-3} a_n z^n}{\Delta^{2g-2}}, \tag{4.10}$$

and to fix it we impose, as usual, gap conditions. The orbifold point, where $t_o = 0$, occurs at $z = \infty$, and we have [6, 32]

$$\mathcal{F}_g(t_o) \sim \frac{2B_{2g}}{2g(2g-2)}\frac{1}{t_o^{2g-2}} + \mathcal{O}(t_o^2). \tag{4.11}$$

Since the expansion contains only even powers of $t_o$, this gives just $g$ conditions. The remaining conditions are obtained by going to the conifold point at $z = 1/16$ and the corresponding conifold frame. The flat coordinate in this frame is given by

$$t_c = \frac{2}{\pi}\int_0^\Delta \frac{K(y)}{1-y}\mathrm{d}y, \tag{4.12}$$

The gap condition in this frame is

$$\mathcal{F}_g(t_c) \sim \frac{B_{2g}}{2g(2g-2)}\left(\frac{2\mathrm{i}}{t_c}\right)^{2g-2} + \mathcal{O}(1). \tag{4.13}$$

This gives $2g - 2$ conditions. Combining the orbifold and the conifold conditions, we get $3g - 2$ conditions in total, which completely fix the holomorphic ambiguity. By using the above ingredients, one can easily compute the $\mathcal{F}_g$s up to very high genus.

## 4.2 Testing the large $N$ instantons

As it was found in [33], in the study of the large order behavior of the genus expansion (4.4) one finds three competing instanton actions. These are given by

$$\mathcal{A}_w = -2\pi i\, t_o, \tag{4.14}$$

$$\mathcal{A}_c = -\frac{1}{4\pi\sqrt{z}} G_{3,3}^{2,3}\left(\begin{matrix} \frac{1}{2}, & \frac{1}{2}, & \frac{1}{2} \\ 0, & 0, & -\frac{1}{2} \end{matrix}\middle| \frac{1}{16z}\right) + \pi^2, \tag{4.15}$$

$$\mathcal{A}_s = \mathcal{A}_c + 2\mathcal{A}_w, \tag{4.16}$$

where $G_{p,q}^{m,n}$ is the Meijer G-function. The first instanton trivially arises from the singular term in (4.11), so we will subtract its effect by removing the polar part in (4.11), as we did in (3.49). The resulting free energies will be denoted as $\mathcal{G}_g$. When we write the instanton actions $\mathcal{A}_c$ and $\mathcal{A}_s$ as in (2.8), in orbifold coordinates, we find $c = 2$. This gives all the ingredients that are needed to compute the instanton amplitudes from (2.9).

We can now check that these instanton amplitudes provide the correct large order behavior of the subtracted free energies $\mathcal{G}_g$. We consider two different cases, $z > 1/16$ and $z < 0$, and avoid the region $0 < z < 1/16$, in which the $\mathcal{F}_g$s acquire an imaginary part. For $z > 1/16$, the closest singularity to the origin of the Borel plane is $\mathcal{A}_c$, which is real. In Fig. 5 we consider

$$z = \frac{i}{15}, \qquad i = 1, \cdots, 20, \tag{4.17}$$

and compare the exact instanton coefficients $\mathcal{F}_n^{(1)}$ with the numerical value extracted from the large order behavior.

Next we consider the case $z < 0$. Now the large order behavior is dominated by the instanton action $\mathcal{A}_s$, which is complex, so we will find an oscillatory asymptotics. In Fig. 6 we plot the coefficients $\widehat{\mathcal{G}}_g(t)$, normalized as in (3.56), as a function of $g$, for different values of $z$. We compare the result to the asymptotic approximation at large $g$, including one, two and three cosine terms of the asymptotic expansion (3.57). We see that, as more terms are included, the approximation becomes better.

In [33], the action $\mathcal{A}_s$ was identified with a D2-brane wrapping a three-cycle in the type IIA string compactification. The expression (2.9), applied to this action, and which we have used to obtain the large genus behavior of the $1/N$ expansion, gives the full quantum amplitude due to this D2-instanton in type IIA theory. It might be possible to test some aspects of this prediction directly in string theory.

## 5 Asymptotics of orbifold Gromov–Witten invariants

### 5.1 Orbifold Gromov–Witten invariants

In topological string theory on a CY manifold, the holomorphic free energies $\mathcal{F}_g(t)$ are generating functions of enumerative invariants. When computed in the large radius frame, they provide conventional Gromov–Witten invariants. If the underlying CY geometry has an orbifold point, there is a corresponding orbifold frame, and the genus $g$ free energies in that frame are generating functionals of orbifold Gromov–Witten invariants. In this section we will focus on a particular example: the CY given by the local $\mathbb{P}^2$ geometry, which can be understood as a resolution of the $\mathbb{C}^3/\mathbb{Z}_3$ orbifold. We will now summarize some basic facts about local $\mathbb{P}^2$ and its orbifold limit, and refer to e.g. [36, 56] for more details.

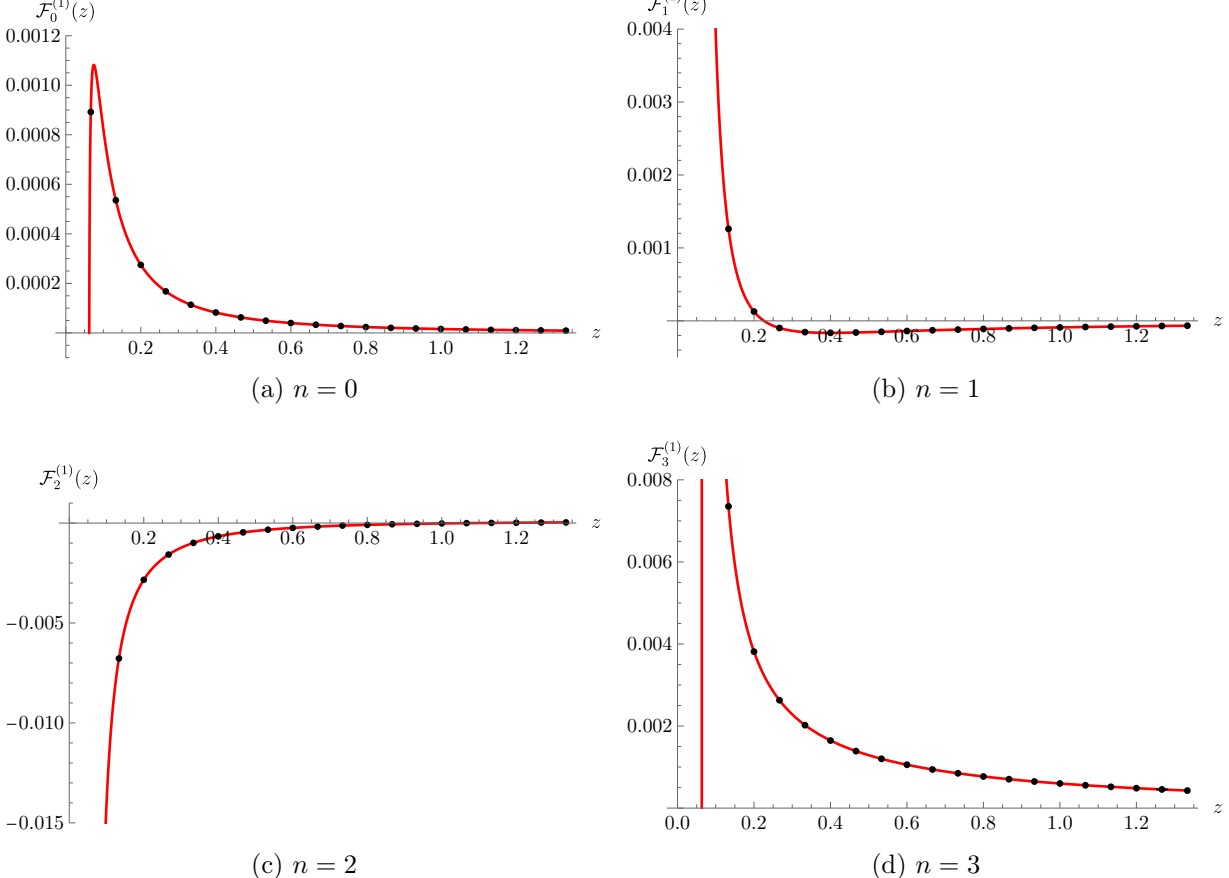

(a) $n = 0$

(b) $n = 1$

(c) $n = 2$

(d) $n = 3$

**Figure 5**: Coefficients $\mathcal{F}_n^{(1)}$, $n = 0, 1, 2, 3$ in the ABJM matrix model, as a function of $z$. The red line is the analytic result extracted from (2.9). The black dots are the numerical approximations extracted from the large order behaviour of the subtracted free energies. For $n = 0$, the relative errors are at most of order $10^{-24}$. For $n = 1$, the relative error is at most of order $10^{-21}$.

The moduli space of local $\mathbb{P}^2$ is parametrized by a complex coordinate $z$. The point $z = 0$ is the large radius point, while at $z = \infty$ one has the orbifold $\mathbb{C}^3/\mathbb{Z}^3$. To parametrize the neighbourhood of the orbifold point it is useful to consider the coordinate $\psi$ defined by

$$\psi^3 = -\frac{1}{27z}. \tag{5.1}$$

The flat coordinate corresponding to the orbifold frame is given by [36, 57]

$$\sigma(z) = 3\psi \, _3F_2\left(\frac{1}{3}, \frac{1}{3}, \frac{1}{3}; \frac{2}{3}, \frac{4}{3} \middle| \psi^3\right). \tag{5.2}$$

The dual coordinate is

$$\sigma_D(z) = -\frac{9}{2} \psi^2 \, _3F_2\left(\frac{2}{3}, \frac{2}{3}, \frac{2}{3}; \frac{4}{3}, \frac{5}{3} \middle| \psi^3\right), \tag{5.3}$$

and it defines a genus zero orbifold free energy, or prepotential, through the relation

$$\sigma_D = 3\frac{\partial \mathcal{F}_0(\sigma)}{\partial \sigma}. \tag{5.4}$$

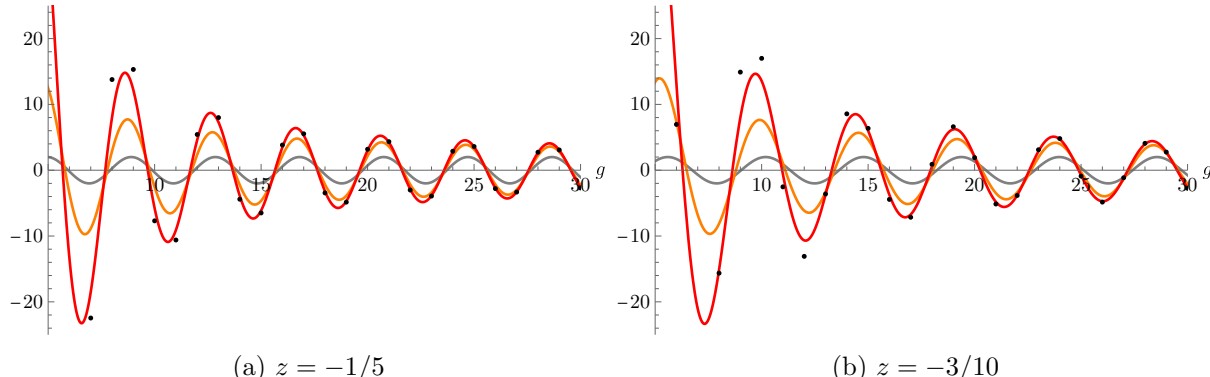

|  (a) $z = -1/5$  |  (b) $z = -3/10$  |

**Figure 6**: Normalized free energies $\widehat{\mathcal{G}}_g(t)$ for the ABJM matrix model (black dots) as compared to the prediction (3.57) for the asymptotics (lines). In grey, we include the leading term; in orange, the subleading term; and, in red, we include the subsubleading term.

The higher genus orbifold free energies $\mathcal{F}_g$ can be computed by using the HAE, since as shown in [9] there are gap conditions which fix the holomorphic ambiguities uniquely. As noted in [36], the $\mathcal{F}_g$s have a series expansion around $\sigma = 0$ in integer powers of

$$\tau = \sigma^3, \tag{5.5}$$

of the form

$$\mathcal{F}_g(\tau) = \sum_{d \geq 0} \frac{\mathcal{N}_{g,d}}{(3d)!} \tau^d. \tag{5.6}$$

We have, for example,

$$\mathcal{F}_0(\tau) = -\frac{\tau}{18} - \frac{\tau^2}{19440} - \frac{\tau^3}{3265920} - \frac{1093\tau^4}{349192166400} - \frac{119401\tau^5}{2859883842816000} + \mathcal{O}\left(\tau^6\right). \tag{5.7}$$

The coefficients $\mathcal{N}_{g,d}$ appearing in this expansion are the orbifold Gromov–Witten invariants of $\mathbb{C}^3/\mathbb{Z}_3$ at genus $g$ and "degree" $d$. In the orbifold theory, $d$ does not refer to a homology class of a curve in the CY target, but indicates that the invariant calculates a correlator of $3d$ twisted fields in the orbifold 2d CFT coupled to gravity. The orbifold Gromov–Witten invariants can be defined independently in algebraic geometry, as integrals over appropriate moduli spaces, and it has been verified that they agree with the results obtained from (5.6) in topological string theory. We refer to [57, 58] for a review and references to the literature.

We note that, in our conventions, we do not include the contribution of constant maps in $\mathcal{F}_g(\tau)$. In particular, the degree zero orbifold GW invariants $\mathcal{N}_{g,0}$ are given by

$$-\frac{1}{2160}, \qquad \frac{1}{544320}, \qquad -\frac{7}{41990400}, \qquad \cdots \tag{5.8}$$

for $g = 2, 3, 4, \cdots$. In contrast, the degree zero invariants calculated in [36, 58] are given by

$$\mathcal{N}_{g,0} + 3\frac{(-1)^{g-1} B_{2g} B_{2g-2}}{4g(2g-2)(2g-2)!}, \tag{5.9}$$

where the second term is the contribution of constant maps. For our asymptotic considerations it is reasonable to define $\mathcal{N}_{g,0}$ as we have done, since the large genus asymptotics of the constant map contributions can be easily worked out in closed form and it is very different from the large genus asymptotics of $\mathcal{N}_{g,0}$.

## 5.2 Asymptotics from instantons

Since the spacetime instantons considered in [20–23] provide the precise large genus asymptotics of the free energies $\mathcal{F}_g$, one could think that they also lead to precise formulae for the asymptotics of the corresponding Gromov–Witten invariants. In the case of conventional Gromov–Witten invariants, this issue was studied in some detail in [34]. The results turn out to be more subtle than expected, however. One finds, for example, that at fixed degree, the conventional Gromov–Witten invariants only grow exponentially with the genus, and precise formulae for this growth can be obtained from the Gopakumar–Vafa invariants [59], without using the asymptotic formulae (2.9), (2.12). This is probably related to the fact that, near the large radius point, the leading Borel singularity is the flat coordinate in the large radius frame, the instanton amplitude is of the form (2.5), and the asymptotics is typically oscillatory [23, 35].

However, in the case of *orbifold* Gromov–Witten invariants, the spacetime instanton amplitudes (2.9), (2.12) give precise predictions for the behavior of $\mathcal{N}_{g,d}$ at fixed $d$ and large $g$. The reason is that, in this case, both the free energies and the instanton amplitudes have a regular expansion around the orbifold point $\sigma = 0$, and one can reorganize the full trans-series in powers of $\tau$. Let us see in detail how this goes.

In order to understand the relevant instantons in the theory, we have to find which are the Borel singularities which are closest to the origin as we approach $\psi \to 0$. To do this, we have generated many $\mathcal{F}_g$s in the orbifold frame and studied numerically the singularities of their Borel transform, by using standard techniques of Padé approximants. For simplicity, we have worked with real negative values of $z$. As a result of this analysis, one finds six singularities, related by conjugation and reflection. The first one occurs at

$$\mathcal{A}_0 = \alpha \frac{\partial \mathcal{F}_0}{\partial \sigma} + \frac{\alpha \beta}{3} \sigma + \mathrm{i}\gamma. \tag{5.10}$$

where[3]

$$\alpha = -\frac{4\pi^2 \mathrm{i}}{\Gamma^3(1/3)}, \qquad \beta = \left(\frac{\Gamma(1/3)}{\Gamma(2/3)}\right)^3, \qquad \gamma = \frac{4\pi^2}{3}. \tag{5.11}$$

We note that $\mathcal{A}_0$ is proportional to the period vanishing at the conifold point at $z = -1/27$, and it is equal to the action $\mathcal{A}_c$ which appeared in the analysis of local $\mathbb{P}^2$ in [22]. As noted in section 2, since $\alpha \neq 0$, the relation (5.10) defines a modified prepotential

$$\mathcal{F}_0^{\mathcal{A}_0} = \mathcal{F}_0 + \frac{\beta}{6}\sigma^2 + \mathrm{i}\frac{\gamma}{\alpha} \tag{5.12}$$

so that

$$\mathcal{A}_0 = \alpha \frac{\partial \mathcal{F}_0^{\mathcal{A}_0}}{\partial \sigma}. \tag{5.13}$$

The other singularities occur at

$$\mathcal{A}_1 = \alpha \mathrm{e}^{-2\pi \mathrm{i}/3} \frac{\partial \mathcal{F}_0}{\partial \sigma} + \frac{\alpha \beta}{3} \mathrm{e}^{-4\pi \mathrm{i}/3} \sigma + \mathrm{i}\gamma,$$
$$\mathcal{A}_2 = \alpha \mathrm{e}^{2\pi \mathrm{i}/3} \frac{\partial \mathcal{F}_0}{\partial \sigma} + \frac{\alpha \beta}{3} \mathrm{e}^{4\pi \mathrm{i}/3} \sigma + \mathrm{i}\gamma, \tag{5.14}$$

and we note that

$$\mathcal{A}_2 = -\overline{\mathcal{A}_1}. \tag{5.15}$$

---

[3]This $\alpha$ should not be confused with the one appearing in (3.22).

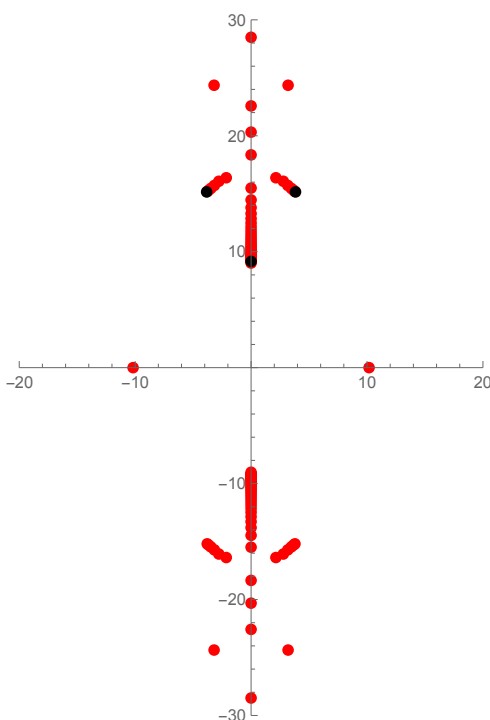

**Figure 7**: Singularities in the Borel plane for $z = -2$, as obtained from the poles of the Borel–Padé transform. The black dot in the positive imaginary axis is $\mathcal{A}_0$, while the two other black dots are $\mathcal{A}_1$ and $\mathcal{A}_2$.

We also have singularities at $-\mathcal{A}_\ell$, $\ell = 0, 1, 2$. A plot of the singularities for $z = -2$ is shown in Fig. 7. We note that, as we go to the orbifold point $\sigma = 0$, the three singularities in the upper half plane coalesce at the value

$$\mathcal{A}_0(\sigma = 0) = \frac{4\pi^2 i}{3}. \tag{5.16}$$

The singularities in the lower half plane coalesce at the conjugate point. In contrast, the large genus asymptotics of the constant map contribution in (5.9) is controlled by an action at $\pm 4\pi^2 i$, which is subleading w.r.t. the singularities $\pm\mathcal{A}_\ell(\sigma = 0)$ considered above. Therefore, although the quantities $\mathcal{N}_{g,0}$ are often combined with the constant map contribution as in (5.9), they have a very different asymptotics at large $g$.

An important symmetry is that

$$\mathcal{A}_1(\sigma) = \mathcal{A}_0\left(e^{2\pi i/3}\sigma\right), \quad \mathcal{A}_2(\sigma) = \mathcal{A}_0\left(e^{4\pi i/3}\sigma\right). \tag{5.17}$$

This says that $\mathcal{A}_{0,1,2}$ form an orbit under the orbifold group $\mathbb{Z}_3$. A similar observation has been made in [23] in the case of the Borel singularities near the orbifold point of the quintic CY. It follows from (5.17) that any symmetric function in the $\mathcal{A}_\ell$, $\ell = 0, 1, 2$, will only contain integer powers of $\tau = \sigma^3$. This will be useful in the following. We also define

$$\begin{aligned}
\mathcal{F}_0^{\mathcal{A}_1} &= \mathcal{F}_0 + \frac{\beta}{6}e^{-2\pi i/3}\sigma^2 + i\frac{\gamma}{\alpha}, \\
\mathcal{F}_0^{\mathcal{A}_2} &= \mathcal{F}_0 + \frac{\beta}{6}e^{2\pi i/3}\sigma^2 + i\frac{\gamma}{\alpha}.
\end{aligned} \tag{5.18}$$

The corresponding instanton amplitudes, obtained from (2.9), will be denoted by $\mathcal{F}_n^{\mathcal{A}_\ell,(1)}$. In order to obtain the asymptotics of $\mathcal{F}_g(\sigma)$, we have to consider the contributions of the three different Borel singularities. Each of them is given by the expression (2.12), and we find in total

$$\mathcal{F}_g(\sigma) \sim \frac{1}{\pi} \sum_{\ell=0}^{2} \sum_{k \geq 0} \mathcal{A}_\ell^{-2g+1+k} \mathcal{F}_k^{\mathcal{A}_\ell,(1)} \Gamma(2g-1-k). \tag{5.19}$$

Due to the $\mathbb{Z}_3$ symmetry, the r.h.s. has a regular expansion in powers of $\tau = \sigma^3$, and by comparing powers of $\tau$ in both sides we can obtain the large genus asymptotics of the orbifold Gromov–Witten invariants at fixed $d$. For example, for the degree zero invariants we find

$$\mathcal{N}_{g,0} \sim \frac{3}{2\pi^2}(-1)^{g-1}\gamma^{-2g+2}\Gamma(2g-1)\exp\left(\frac{\alpha^2\beta}{6}\right)\left\{1 + \frac{18 - 6\alpha^2\beta + \mathrm{i}\alpha^3\gamma}{18}\frac{1}{2g} + \cdots\right\}, \tag{5.20}$$

while for the degree one invariants we obtain

$$\frac{\mathcal{N}_{g,1}}{3!} \sim \frac{3}{2\pi^2}(-1)^g\gamma^{-2g}(2g)^3\Gamma(2g-1)\frac{\mathrm{i}\alpha^3\beta^3}{162\gamma}\exp\left(\frac{\alpha^2\beta}{6}\right)\left\{1 + \mathcal{O}\left(g^{-1}\right)\right\}. \tag{5.21}$$

Note that, since $\alpha$ is purely imaginary, the r.h.s of the above asymptotic equalities is real, as it should be. It is straightforward to extend these formulae to all orders in $1/g$, by simply considering higher order corrections in $g_s$ in the instanton amplitudes. Similarly, we can obtain results for all degrees $d$ by simply expanding the r.h.s. of (5.19) in powers of $\tau$.

We have explicitly verified many of these instanton predictions by studying the large genus asymptotics of the invariants $\mathcal{N}_{g,d}$, for different values of $d$. Let us mention two of these two checks, for $d = 0$ and $d = 1$. The sequence

$$2g\left\{\frac{\mathcal{N}_{g,0}}{(-1)^{g-1}\gamma^{-2g+2}\Gamma(2g-1)} - \frac{3}{2\pi^2}\exp\left(\frac{\alpha^2\beta}{6}\right)\right\} \tag{5.22}$$

should asymptote the number

$$\frac{3}{2\pi^2}\exp\left(\frac{\alpha^2\beta}{6}\right)\frac{18 - 6\alpha^2\beta + \mathrm{i}\alpha^3\gamma}{18} = \frac{3}{2\pi^2}\mathrm{e}^{-\sqrt{3}\pi}\left(1 + 2\sqrt{3}\pi - \frac{128\pi^8}{27\Gamma^9(1/3)}\right) \approx 0.0036573... \tag{5.23}$$

Similarly, the sequence

$$\frac{\mathcal{N}_{g,1}}{(-1)^{g-1}\gamma^{-2g+2}(2g)^3\Gamma(2g-1)} \tag{5.24}$$

should asymptote the number

$$\frac{3}{2\pi^2}\frac{\mathrm{i}\alpha^3\beta^3}{162\gamma}\exp\left(\frac{\alpha^2\beta}{6}\right) = \frac{3}{2\pi^2}\mathrm{e}^{-\sqrt{3}\pi}\frac{5832\pi^4}{\Gamma^9(-1/3)} \approx -0.00124176... \tag{5.25}$$

We plot these sequences, up to $g = 39$, together with their second Richardson transform, in Fig. 8. By using further transforms we can match the theoretical predictions with a relative error of $10^{-11}$.

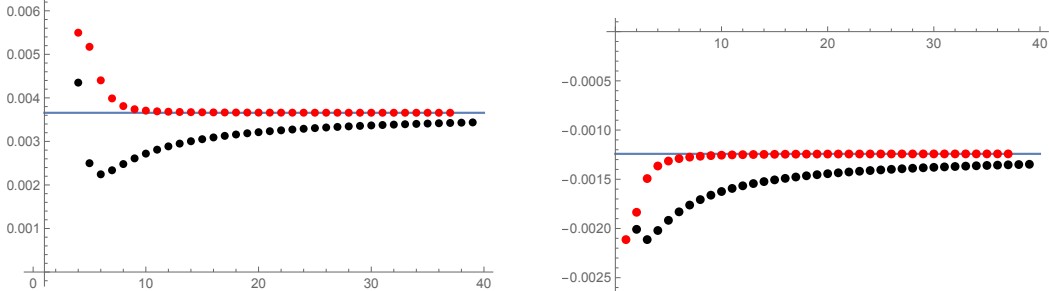

**Figure 8**: On the left, the sequence (5.22) and its second Richardson transform (black and red dots, respectively), as compared to its predicted asymptotic limit (5.23) (blue line). On the right, the sequence (5.24) and its second Richardson transform (black and red dots, respectively), as compare to its asymptotic limit (5.25) (blue line).

## 6 Conclusions

In this paper we have shown that the instanton amplitudes for topological strings obtained in [20–23] give the correct non-perturbative corrections due to large $N$ instantons in Hermitian matrix models. Our results solve in part the puzzle raised in [30]. In that paper it was checked that, in the two-cut cubic matrix model with $\alpha = 0$, the large genus asymptotics of the $\mathcal{F}_g$s was controlled by the dual instanton action (3.50). However, the subleading coefficients appearing in the asymptotic formula (2.12) were not known explicitly. A naif eigenvalue tunneling analysis suggests that the instanton amplitude is given, in the one-modulus case, by an expression of the form (see e.g. [28])

$$\exp\left[\mathcal{F}(t - cg_s, g_s) - \mathcal{F}(t, g_s)\right]. \tag{6.1}$$

This *does* not lead to the correct asymptotics, as it was noted in [30]. In view of the results of this paper, it is clear that the expression (6.1) is missing the non-trivial prefactor appearing in (2.9). From the point of view of [20–23], the problem with (6.1) is that it does not satisfy the appropriate boundary conditions due to the gap behavior (3.19).

What we are still lacking is a microscopic derivation of (2.7) and (2.9) from the dynamics of the matrix model eigenvalues, in the same way that (6.1) is explained by eigenvalue tunneling. In [30] it was suggested, based on the results of [60], that to go beyond (6.1) one has to take into account a new type of instanton. This new instanton has found an eigenvalue description very recently [61] in terms of super matrix models (see [62] for its applications), and this makes it possible to provide a rationale for (2.9) in terms of eigenvalue instantons and "anti-eigenvalue" instantons [63].

In this paper we have addressed very simple aspects of the full resurgent structure of the $1/N$ expansion of matrix models. The conjectures of [22, 23] give information about e.g. multi-instanton amplitudes, and we have verified some of them, but more work remains to be done in this direction. We also note that the conjectures of [22, 23] do not give detailed information on the structure of Borel singularities and on the Stokes constants. We expect the resurgent structure of matrix models with polynomial potentials to be simpler than in the case of topological string theory on toric or compact CY threefolds, and perhaps one can find a complete description of these missing ingredients.

As we have seen in this work, the large $N$ instantons of the ABJM matrix model are also described by the topological string instanton amplitudes. This is perhaps not so surprising, since

the $1/N$ expansion of the ABJM matrix model coincides with the genus expansion of topological string theory on the local $\mathbb{F}_0$ geometry [5, 6]. There is another class of non-conventional matrix models, associated to quantum mirror curves [54, 64], whose large $N$ instantons are described by (2.7), due essentially the same reasons; namely, their $1/N$ expansion is conjectured to be given by the genus expansion of a topological string. In all these cases, we are lacking a microscopic picture of the large $N$ instantons in the matrix models themselves. It would be also interesting to see whether the large $N$ instantons of the matrix models appearing more generally in the localization of Chern–Simons–matter theories are also described by (2.7).

Another interesting question is the following. It was found in [65] that the Borel resummation of the $1/N$ expansion of the ABJM matrix model is not enough to reproduce its exact value, and non-perturbative corrections are needed. It is likely that the large $N$ instantons of the ABJM matrix model described in this paper provide the sought-for non-perturbative corrections. Eventually, one would like to have a complete "semiclassical decoding" of the exact matrix model in terms of a Borel resummed trans-series. Some first steps in this decoding were achieved in [35] for a close cousin of the ABJM matrix model, namely the local $\mathbb{P}^2$ matrix model introduced in [64], but much remains to be understood.

Finally, we note that the results we have obtained for the asymptotics of orbifold Gromov–Witten invariants in $\mathbb{C}^3/\mathbb{Z}_3$ are perhaps the simplest ones that can be derived from the topological string instanton amplitudes (2.7). They give new results in Gromov–Witten theory and provide at the same time precision tests of the instanton amplitudes. It would be interesting to generalize these results to other Calabi–Yau orbifold points, both in the toric and the compact cases.

## Acknowledgements

We would like to thank Jie Gu, Rahul Pandharipande, Ricardo Schiappa and Max Schwick for useful comments and discussions. Thanks in particular to Ricardo Schiappa and Jie Gu for his comments on the draft version of this paper. This work has been supported in part by the ERC-SyG project "Recursive and Exact New Quantum Theory" (ReNewQuantum), which received funding from the European Research Council (ERC) under the European Union's Horizon 2020 research and innovation program, grant agreement No. 810573.

## A A useful parametrization of the cubic matrix model

In this Appendix we review the parametrization of the two-cut cubic matrix model which we use to fix the holomorphic ambiguities.

One problem of the parameters $z$, $\alpha$ appearing in the spectral curve (3.22) is that the roots $x_i$ have very complicated expressions in terms of them. It is therefore useful to introduce some intermediate parameters $z_{1,2}$, first considered in [47]. They are defined by

$$
\begin{aligned}
\frac{1}{4}(x_2 - x_1)^2 &= z_2, \\
\frac{1}{4}(x_4 - x_3)^2 &= z_1, \\
x_1 + x_2 + x_3 + x_4 &= 0, \\
\frac{1}{4}\left[(x_3 + x_4) - (x_1 + x_2)\right]^2 &= 4 - 2(z_1 + z_2).
\end{aligned}
\tag{A.1}
$$

The modulus $z$ and parameter $\alpha$ are then given by:

$$z = \frac{1}{4}\left(8(z_1 + z_2) - 3(z_1^2 + z_2^2) - 10z_1z_2\right),$$

$$\alpha = 2(z_2 - z_1)\sqrt{1 - \frac{z_1 + z_2}{2}}. \tag{A.2}$$

The periods $t_{1,2}$ can be calculated in a power series around $z_1 = z_2 = 0$ [47], and one finds

$$t_1 = \frac{z_1 I}{4} - \frac{z_1 z_2}{2I}K(z_1, z_2, I),$$

$$t_2 = -\frac{z_2 I}{4} + \frac{z_1 z_2}{2I}K(z_1, z_2, I), \tag{A.3}$$

where [49]

$$K(z_1, z_2, I) = \sum_{m,n \geq 0} \frac{2^{-2m-2n-1}(m+n)\Gamma(2m+2n)}{\Gamma(m+1)\Gamma(m+2)\Gamma(n+1)\Gamma(n+2)} \frac{z_1^n z_2^m}{I^{2(n+m)}} \tag{A.4}$$

and

$$I = 2\sqrt{1 - \frac{z_1 + z_2}{2}}. \tag{A.5}$$

We note that the point $t_1 = t_2 = 0$ where we implement the gap condition (3.41) corresponds to $z_1 = z_2 = 0$.

It is convenient to find a formula for the holomorphic propagator as a function of $z_{1,2}$ which allows us to make fast expansions around $z_1 = z_2 = 0$. Let us introduce the functions

$$\lambda = 4z_1z_2, \qquad a = 4 - 3(z_1 + z_2), \tag{A.6}$$

as well as the elliptic modulus

$$k_1^2 = \frac{\lambda}{\left(a + \sqrt{a^2 - \lambda}\right)^2}, \tag{A.7}$$

which is analytic at $z_1 = z_2 = 0$. Then, one finds

$$\mathcal{S} = \sigma(z_1, z_2) - \delta(z_1, z_2)\left\{\frac{a + \sqrt{a^2 - \lambda}}{a^2 - \lambda}\frac{E(k_1^2)}{K(k_1^2)} - \frac{1}{\sqrt{a^2 - \lambda}}\right\}, \tag{A.8}$$

where

$$\sigma(z_1, z_2) = \frac{1}{2}\left(32 - 24(z_1 + z_2) + 3(z_1^2 + z_2^2) + 10z_1z_2\right), \tag{A.9}$$

and

$$\delta(z_1, z_2) = 4(a^2 - \lambda). \tag{A.10}$$

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
