# Peer review of "Large N instantons from topological strings"

_SciPost Physics_

## Round 1 · Referee Report · Min-xin Huang (Referee 1) · 2024-4-28

Strengths

  1. Non-perturbative formulation is probably the most outstanding problem in string/M-theory. Non-perturbative effects are also important in many topics in mathematics and physics, in particular in situations involving strong coupling dynamics. The author’s approach is a promising direction where many concrete precise results can be obtained.
  2. The numerical tests have some quite impressive precision.
  3. These examples establish the universality of the general formalism proposed in the previous papers.

Weaknesses

  1. The main novelty of general formalism has been proposed in the previous papers. The current manuscript is just an application to more though still non-trivial examples.
  2. The current tests are mostly numerical. It would be better have some analytic derivation or proof of some of the results.

Report

The authors apply the results in the previous recent papers [22, 23] to some more non-trivial examples. In the previous papers, in the context of topological string theory on Calabi-Yau 3-folds, some novel conjectures were proposed concerning the relation between the instanton action and the periods of Calabi-Yau geometry, as well as the formulas for non-perturbative trans-series free energy. The manuscript summarized the developments in Section 2, e.g. formulas (2.4), (2.9), then test the formula (2.9) in three more examples related to matrix models using the large order asymptotics of the perturbative free energy. Such tests often involve some heroic efforts of computations to very large order.

As in the previous papers, the manuscript provides some impressive numerical tests. As can be seen in the various figures, the dots extracted from the large order asymptotics lie well on the lines from the proposed trans-series formula, providing convincing evidence for its validity. We should note that the computations of perturbative free energy are by themselves quite non-trivial, as one needs to use various boundary conditions to fix the holomorphic ambiguity. But a surprising aspect in [22, 23] is that once the perturbative part is determined, the boundary conditions for trans-series parts are relatively quite simple. Using previous works on the holomorphic anomaly equation for the trans-series parts, simple formulas at least for low orders can be written, as e.g. in (2.9).

I think the manuscript contains some solid research results. I recommend publication of the manuscript.

Recommendation

Publish (easily meets expectations and criteria for this Journal; among top 50%)

---

## Round 1 · Referee Report · Anonymous (Referee 2) · 2024-5-11

Strengths

Numerical tests of non-perturbative correction for various matrix models are very non-trivial.

Report

The non-perturbative correction of matrix models is an important subject. The authors use results from topological string instanton amplitudes to study these corrections for generic, multi-cut, hermitian matrix models. Following previous works, the paper proposes an explicit form and tests it carefully for two-cut solutions of a cubic matrix model. The numerical agreement is very impressive. They also test for ABJM theory, whose non-perturbative correction is interpreted as D2-instantons in previous works. It is interesting to establish a closed form of non-perturbative correction for matrix models from topological string instanton amplitudes.
Relating to the results in this paper, for the ABJM theory, non-perturbative correction for the matrix model was also studied in Fermi gas formalism initiated by one of the authors. There, another expansion for D2-instantons was known and the final result on non-perturbative corrections was also given by topological string theory. The relation to the present work is not very clear to me. Do the result from Fermi gas and the current proposal relate to each other in a symplectic transformation? The expansion in Fermi gas formalism seems to work only for ABJM-like theories, but the proposal in the present paper seems more general. I would suggest to provide some comments on the relations, if the authors have some in mind.
Namely, I think that the results of the manuscript is very impressive and I recommend the manuscript for publication after adding some comments.

Requested changes

Comments on the relation to previous results in Fermi gas formalism.

Recommendation

Ask for minor revision

  • validity: -
  • significance: -
  • originality: -
  • clarity: -
  • formatting: -
  • grammar: -

Author:  Ramon Miravitllas  on 2024-05-20  [id 4494]

(in reply to Report 2 on 2024-05-11)

We thank the referee for their review and comments.
We have updated the manuscript with a few comments about the Fermi gas formalism and the relation with our results (see the last paragraph in section 4.2 and also the updated Conclusions).

---

## Editorial Decision

resubmitted